# Impact of COVID-19 epidemic curtailment strategies in selected Indian states: An analysis by reproduction number and doubling time with incidence modelling

Arun Mitra[1,‡], Abhijit P. Pakhare[2], Adrija Roy[3], Ankur Joshi[4]*

1 Department of Community and Family Medicine, All India Institute of Medical Sciences, Bhopal, India,
2 Department of Community and Family Medicine, All India Institute of Medical Sciences, Bhopal, India,
3 Community Medicine and Family Medicine, All India Institute of Medical Sciences, Bhubaneshwar, India,
4 Department of Community and Family Medicine, All India Institute of Medical Sciences, Bhopal, India

☯ These authors contributed equally to this work.
‡ Independent Researcher.
* ankur.cfm@aiimsbhopal.edu.in

**Data Availability Statement:** All data is available through API of crowdsourced database. Code for fetching data is available in Supplementary File. A copy of the cleaned data has been been provided in

## Abstract

The Government of India in-network with the state governments has implemented the epidemic curtailment strategies inclusive of case-isolation, quarantine and lockdown in response to ongoing novel coronavirus (COVID-19) outbreak. In this manuscript, we attempt to estimate the impact of these steps across ten selected Indian states using crowd-sourced data. The trajectory of the outbreak was parameterized by the reproduction number ($R_0$), doubling time, and growth rate. These parameters were estimated at two time-periods after the enforcement of the lockdown on 24th March 2020, i.e. 15 days into lockdown and 30 days into lockdown. The authors used a crowd sourced database which is available in the public domain. After preparing the data for analysis, $R_0$ was estimated using maximum likelihood (ML) method which is based on the expectation minimum algorithm where the distribution probability of secondary cases is maximized using the serial interval discretization. The doubling time and growth rate were estimated by the natural log transformation of the exponential growth equation. The overall analysis shows decreasing trends in time-varying reproduction numbers ($R_{(t)}$) and growth rate (with a few exceptions) and increasing trends in doubling time. The curtailment strategies employed by the Indian government seem to be effective in reducing the transmission parameters of the COVID-19 epidemic. The estimated $R_{(t)}$ are still above the threshold of 1, and the resultant absolute case numbers show an increase with time. Future curtailment and mitigation strategies thus may take into account these findings while formulating further course of action.

the Supporting Information for reproducibility of
the R Code.

**Funding:** The author(s) received no specific
funding for this work.

**Competing interests:** The authors have declared
that no competing interests exist.

## Introduction

The World Health Organization (WHO) declared the Novel Coronavirus Outbreak (COVID-19) as a pandemic on 11[th] March 2020, calling for immediate action to be taken on by all countries in terms of stepping up treatment, detection, and reduction of transmission. As of 26[th] April 2020, a total of 2.96 million confirmed cases with over 200 thousand deaths reported in 185 countries [1]. The Ministry of Health and Family Welfare, Govt. of India reported over 20000 cases across 32 states/union territories with 872 deaths [2]. Government of India initiated various non-pharmaceutical interventions which include social distancing measures like lockdown. The nationwide lockdown was enforced in India on 24[th] March 2020 resulting in restrictions on unnecessary travel, closure of schools, colleges, and the prohibition of mass gatherings. Despite an assumed uniform susceptibility of the Indian population to COVID-19, the trends till now are showing a variegated force of infection in different states. It is important to capture these regional and state-specific variations as they may offer crucial insights into the current mitigation strategies. The quantification of this variation may aid in planning future intervention strategies and be vital to understand the impact of the lockdown strategy adopted by the country to curtail the impact and flatten the peak of the COVID-19 epidemic [3–5]. The scope of this manuscript is to estimate the time-varying reproduction number ($R_{(t)}$) and doubling time before the commencement of lockdown, 15 days into the lockdown (early epidemic) and at day-30 of the lockdown to see the cumulative effect of curtailment strategies (inclusive of lockdown) in selected states. The ten states reporting the highest numbers of COVID-19 cases as on 23[rd] April 2020 were chosen for this analysis. The database used for the analysis is in open-domain at www.covid19india.org. $R_{(t)}$ and doubling time were chosen for their primary role in reflecting the force, consistency and continuity of an infectious disease which is critically important in COVID-19.

## Methodology

### Data source

COVID-19 cases, deaths and recoveries in India are reported by state public health agencies to the Ministry of Health and Family Welfare (MoHFW), Government of India. The MoHFW releases the testing guidelines and amends them as per the epidemiological scenarios and expert opinion. The eligibility for testing includes patients presenting with suspected symptoms in hospitals, exposed healthcare workers as well as contacts identified through contact tracing. All the hospitals or outreach services notifies cases to district level public health authority which in turn compiles data and reports to state level public health authority. Daily report on number of cases, recoveries and deaths along with the line-list of cases are sent from states to MoHFW on a dedicated portal. State public health authorities simultaneously publishes daily bulletin of same reports.

The data source used for this study is compiled from these state bulletins, official handles of state governments, and health ministries and maintained at www.covid19india.org [6]. This crowd-sourced database and website is maintained by a group of volunteers who curate and verify the data coming from several sources mentioned above. It is validated, updated periodically and published into an application programming interface (API) and Google spreadsheet which is accessible at api.covid19india.org for the public. Apart from the patient-level data, the API includes district-level, state-level, and national-level datasets. We used the data from the line-listing of the cases reported as positive for COVID-19. The data was iteratively and progressively accessed through the database in coherence with creation and improvement in analysis code. The last access to the database was made on 1[st] May 2020. We truncated the data up

to 23rd April 2020 for this study. This buffer period of 7 days offered some immunity against the possible delay to add the cases and our limitation to access the data in real-time.

## Data preparation

The data were prepared for analysis in the following steps:

1. Loading the *.json file containing the raw line-list data

2. A data-frame is then created and variables of interest are selected

3. The imported cases are then coded in the following fashion:

   a. All cases with travel history outside the country before the lockdown are coded as imported cases. These cases were removed from further analysis.

   b. All cases reported after 15 days of the lockdown (i.e. 9th April 2020) irrespective of their travel history are coded as local cases

4. Data from the top 10 states with highest number of cases were subset.

Respective incidence objects were created by adding number of local cases reported on each date based on the timeframes described below.

We divided the timeline of the epidemic into three phases. The first phase was before lockdown i.e. 25th March 2020, the second phase was the early epidemic phase (15 days into the lockdown), and the third phase was till day-30 of the lockdown. However, the transmission parameters before lockdown were not estimated due to certain considerations described in the supporting information (S1 Table). The second phase (15 days into the lockdown) was considered as the baseline for estimation of the transmission parameters.

## Data analysis

Statistical software R, version 3.6.2 was used to perform all statistical analysis and model development [7]. We used the package "*incidence*" [8, 9] to model the incidence and estimate growth rate and doubling time, and package "*R0*" [10] to estimate the time-varying reproduction number ($R_{(t)}$) for different states. The growth rate and doubling time were estimated using the "fit()" function of the "*incidence*" package fits an exponential model to the incidence data in the form of: $log(y) = r * t$ + b; where y is the incidence, t is the time (in days) and r is the growth rate while b is the intercept or origin. The doubling time is then estimated by dividing the natural logarithm of 2 with the growth rate of the epidemic i.e. doubling time (d) = $log(2)$ / r. The package "*projections*" was used to simulate the epidemic outbreaks and project their respective trajectories based on the state-specific transmission parameters [11]. Detailed description of the methods employed have been submitted in the supporting information. The computational work-flow of the analyses performed along with the R code has been submitted at www.protocols.io [12].

## Estimation of reproduction number

The time-varying reproduction number ($R_{(t)}$) was estimated by using the maximum likelihood (ML) method [13]. This method presumes that all the secondary cases linked to the primary cases follow a Poisson process (event rate is constant), and the corresponding serial interval follows a multinomial distribution. This leads to a gradual trend towards zero secondary cases (as time progresses) arising from primary cases during a specific time-step. The gradient depends on the probability density function (PDF) of the serial interval. The "est R0.ML()"

function in the "*R0*" package was used for the estimation of $R_{(t)}$. This runs an expectation maximum (EM) algorithm, which maximizes the distribution probability of primary and secondary cases with reference to time. This method assumes that infectee always develops symptoms only after infector; thus, the value of the serial interval will be positive.

### Serial interval

For the probability density function (PDF), we could not obtain the generation time (time lag between the infection in the primary case and secondary cases) distribution directly with infector-infectee pairs due to the lack of data availability. Therefore, it was substituted with the serial interval distribution discretized on a 1-day time-step. This was created using the generation.time()" function in the "*R0*" package. For parametrization purposes, we chose a gamma distribution as it accommodates for the underlying changing number of events in the constant event rate (Poisson process). The distribution assumptions were aligned with the emerging literature as well as the observed plausible transmission dynamics. The mean and standard deviation for serial interval approximations was 4.4 days and 3 days, respectively [14]. The shape (number of events in time step) and scale (the reciprocal of event rate) of the distribution were 2.15 and 2.04 respectively.

### Modelling incidence and projections

Regression of log-incidence over time was used to model the cumulative-incidence. The package "*projections*" was used to simulate 1000 probable epidemic outbreak trajectories and plot the future daily cumulative incidence predictions based on probability mass function dependent branching process assuming it follows a Poisson distribution [15]. This was done to curve-fit the robustness of *R(t)* and check it by plotting against new incidence. The reproduction numbers of the third phase (i.e. 30 days into lockdown) were used to model the incidence and predict the cumulative caseload for the selected states.

### Ethical issues

Dataset used in this study was generated by using state bulletins or official handles of concerned states and does not contain any identifiers. The study did not involve an interview or questionnaire and did not require the patient's consent and Ethics Committee approval.

## Results

A total of 23,040 COVID-19 cases have been reported in India as of 23[rd] April 2020 of which 20,590 cases (89.4%) were seen in the selected 10 states. The proportion of imported cases was less than 2% in all the 10 states.

### Demographics

Table 1 shows the demographics and key relevant statistics pertaining to COVID-19 epidemic of the chosen states (as of 23[rd] April 2020).

As shown in Fig 1, the composite plot where lines diagram represents the trends in cumulative number of cases with reference to time and the bars show the proportional increase in cases per day for a specific state on that specific day. The two vertical lines divide the whole interface into before lockdown, early epidemic (15 days into lockdown) and current time frame (30 days into lockdown).

Table 1. Key relevant statistics pertaining to COVID-19 epidemic and demographics of the chosen states (as of 23$^{rd}$ April 2020).

| State Name | Population (in Million)$^{\#}$ | Cases | Deaths | Recovered | CFR | Recovery Rate | Infection rate† | Tests performed† | Positivity Rate |
|---|---|---|---|---|---|---|---|---|---|
| *Maharashtra* | 112.4 | 6427 | 282 | 840 | 4.39 | 13.07 | 57.18 | 794 | 7.21 |
| *Gujarat* | 60.4 | 2624 | 112 | 252 | 4.27 | 9.60 | 43.44 | 702 | 6.19 |
| *Delhi* | 16.8 | 2376 | 50 | 808 | 2.1 | 34.01 | 141.43 | 1819 | 7.77 |
| *Rajasthan* | 68.5 | 1964 | 28 | 451 | 1.43 | 22.96 | 28.67 | 1018 | 2.82 |
| *Madhya Pradesh* | 72.6 | 1687 | 93 | 203 | 5.51 | 12.03 | 23.24 | 338 | 6.87 |
| *Tamil Nadu* | 72.1 | 1683 | 20 | 752 | 1.19 | 44.68 | 23.34 | 915 | 2.55 |
| *Uttar Pradesh* | 199.8 | 1510 | 24 | 206 | 1.59 | 13.64 | 7.56 | 228 | 3.32 |
| *Telangana* | 35.2 | 970 | 25 | 252 | 2.58 | 25.98 | 27.56 | 425* | 6.48* |
| *Andhra Pradesh* | 49.5 | 893 | 27 | 141 | 3.02 | 15.79 | 18.04 | 970 | 1.86 |
| *West Bengal* | 91.3 | 456 | 15 | 79 | 3.29 | 17.32 | 4.99 | 88 | 5.71 |

# According to Census 2011.

† per million.

*Testing data for 19$^{th}$ April 2020 was used.

CFR–Case Fatality Rate.

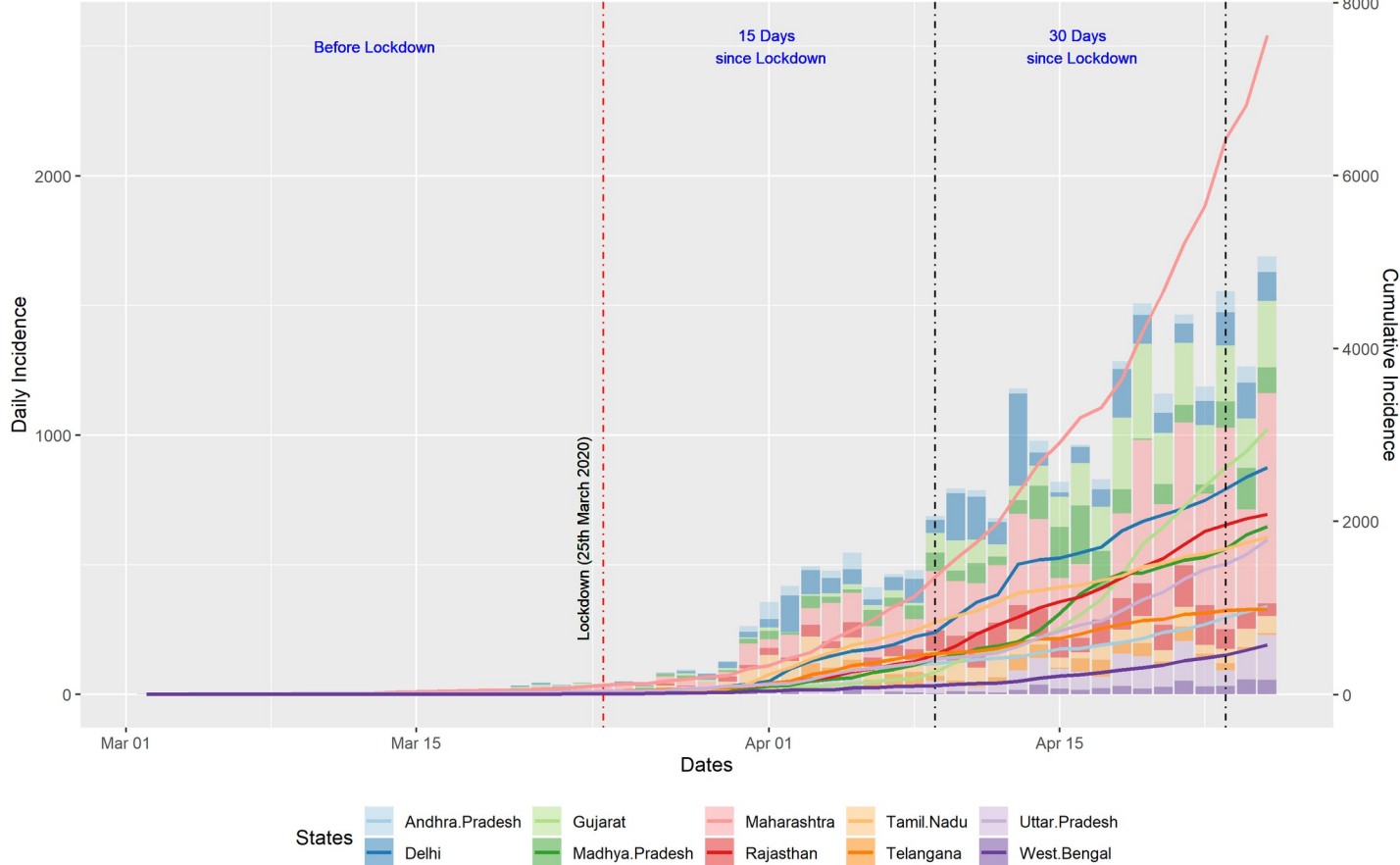

**Fig 1. Composite plot of daily and cumulative incidence of COVID-19.** The daily new cases (daily incidence) of the selected states are represented on the primary y-axis as columns. The lines on the secondary y-axis represent the total cumulative cases (cumulative incidence). The three vertical lines on the x-axis represent the three time-points considered for the study. The first vertical line represents the initiation of lockdown; the second vertical line represents the period of 15 days into lockdown, whereas the third vertical line represents 30 days into lockdown.

**Table 2. Estimates of the epidemiological parameters of the chosen states at different time-points of lockdown (LD) (as of 23rd April 2020).**

| State | Reproduction Number | | Doubling Time | | Growth Rate | |
|---|---|---|---|---|---|---|
| | *15 days of LD* | *30 days of LD* | *15 days of LD* | *30 days of LD* | *15 days of LD* | *30 days of LD* |
| *Maharashtra* | 1.93 [1.77–2.11] | 1.54 [1.49–1.59] | 4.91 [4.17–5.97] | 5.2 [4.76–5.74] | 0.14 [0.12–0.17] | 0.13 [0.12–0.15] |
| *Gujarat* | 1.72 [1.38–2.11] | 2.05 [1.91–2.18] | 10.08 [5.61–49.83] | 4.79 [4.11–5.75] | 0.07 [0.01–0.12] | 0.14 [0.12–0.17] |
| *Delhi* | 3.64 [3.08–4.26] | 1.9 [1.77–2.04] | 4.91 [4–6.35] | 5.84 [5.02–6.96] | 0.14 [0.11–0.17] | 0.12 [0.1–0.14] |
| *Rajasthan* | 2.19 [1.83–2.58] | 1.44 [1.35–1.54] | 5.78 [4.87–7.09] | 5.98 [5.39–6.72] | 0.12 [0.1–0.14] | 0.12 [0.1–0.13] |
| *Madhya Pradesh* | 2.14 [1.79–2.53] | 1.94 [1.78–2.1] | 4.06 [3.04–6.1] | 6.61 [5.02–9.67] | 0.17 [0.11–0.23] | 0.10 [0.07–0.14] |
| *Tamil Nadu* | 4.62 [3.83–5.51] | 3.99 [3.31–4.77] | 3.64 [2.94–4.78] | 6.75 [5.31–9.25] | 0.19 [0.15–0.24] | 0.10 [0.07–0.13] |
| *Uttar Pradesh* | 2.2 [1.82–2.62] | 1.52 [1.41–1.64] | 6.93 [5.3–10.04] | 6.78 [5.9–7.98] | 0.10 [0.07–0.13] | 0.10 [0.09–0.12] |
| *Telangana* | 2.55 [2.11–3.05] | 2.41 [1.99–2.88] | 4.9 [4.01–6.3] | 8.07 [6.5–10.63] | 0.14 [0.11–0.17] | 0.09 [0.07–0.11] |
| *Andhra Pradesh* | 5.72 [4.34–7.37] | 1.37 [1.25–1.5] | 3.76 [2.79–5.75] | 6.13 [4.92–8.11] | 0.18 [0.12–0.25] | 0.11 [0.09–0.14] |
| *West Bengal* | 2.05 [1.48–2.76] | 1.56 [1.35–1.79] | 5.38 [3.56–11.05] | 7.03 [5.79–8.94] | 0.13 [0.06–0.19] | 0.10 [0.08–0.12] |

The numbers in the square brackets represent the 95% confidence intervals.

## Epidemiological parameters

Table 2 shows the effective reproduction number ($R_{(t)}$) at 15 days and 30 days into lockdown. The respective doubling time is also shown at these time points. The estimates in doubling time during the early epidemic in some states show a high degree of unreliability with wide confidence intervals. Doubling time also changed with the evolving outbreak. Increase in doubling time means a slow growth rate of an outbreak. Five states reported an increase in doubling time, and four states reported negligible change in doubling time. The state of Gujarat reported a decrease in doubling time which could mean that there is no slowdown of the outbreak. Seven of the ten selected states saw a reduction in reproduction number ($R_{(t)}$) between the early epidemic phase and the current timeframe. The highest decrease in $R_{(t)}$ was seen in Andhra Pradesh (73%) followed by Delhi (43%) and Rajasthan (30%). Telangana and Tamil Nadu saw stable $R_{(t)}$ during this time period while Gujarat, on the other hand, saw an increase. The growth rates of 8 of 10 states showed a decline between the two time intervals. Uttar Pradesh did not show a decline in growth rate, whereas Gujarat showed an increase. Additional analysis is provided in the S1 Appendix along with the R Code.

## Modelling incidence & future projections

Amongst the 10-day projected cases, seven of the ten states had observed values within the predicted range. States of Rajasthan, Madhya Pradesh and Telangana observed lesser cases than predicted (S2 Table). A detailed description of the methods of projections is provided in the S2 Appendix.

## Discussion

This study evaluates the impact of nationwide lockdown on COVID-19 cases in ten states of India. At the beginning of the outbreak, states reported high transmissibility and low doubling time. The nationwide lockdown was implemented from 24th March 2020. The time-varying reproduction number ($R_{(t)}$) in several states has come down by the adopted curtailment strategies, including lockdown compared to what was estimated at the beginning of the epidemic. As the final epidemic size's relation with $R_{(t)}$ is exponential and not linear, this reduction if sustained, may considerably decrease the total number of affected persons compared to initial estimates. However, two factors should be considered at this moment. Firstly, the $R_{(t)}$ needs to

be further reduced in-order to flatten or change the trajectory of the epidemic curve, and the one may perceive the state-wise variations in its magnitude. Secondly, although the doubling time has increased in relative terms, the epidemic still follows an exponential trajectory, and the current daily incidence is much more as compared to the beginning of the epidemic. Our results are similar to the work done by Sam Abbott and colleagues, where they estimated the time-varying reproduction number of COVID-19 in select Indian states [16, 17].

There are several approaches for $R_{(t)}$ estimations like exponential growth (EG), sequential bayesian (SB), and time-dependent (TD) approach apart from the maximum likelihood (ML) approach used here [18–20]. The SB method requires a prior distribution of gamma for calculating the posterior distribution. It is better suited for the initial stage of the epidemic (exponential phase), where there is no intervention like quarantine, isolation, vaccine etc. We faced an empirical issue while attempting this method due to the erratic trends incidence of cases in several states. (many zero-incidence days following a non-zero incidence day). Thus, for estimating the prior β/effective contact distribution, the reported initial cases occurring before the last zero-incidence day had to be removed. In some states where there was misreporting of cases during the early epidemic, this removal constituted a significant proportion of the cases resulting in an $R_{(t)}$ that was zero. In-depth literature review suggests that during the initial phase of an epidemic (functioning cut off for initial phase is sometimes also reported as the square root of the susceptible fraction) ML method produces comparable results to that of SB method with a capacity to accommodate these erratic trends by minimizing prior values. The EG method computes the $R_{(t)}$ using the Poisson regression for early exponential growth period of an epidemic. However, this method has been criticized for precipitating several biases and violating assumptions [20]. There is an innate subjectivity component in the theoretical aspect of the exponential growth method despite the proposed goodness of fit statistic and deviance R square measures and potential under-reporting and asymptomatic cases in the context of COVID-19 may make matters worse. Moreover, as the purpose of this investigation was to measure the difference in initial 15 and later 15 days of lockdown, the $R_{(t)}$ estimated for the initial 15 days by the exponential method may be an overestimate because of better-fitting thus incorporating an inherent confounding [21]. The TD method calculates the reproduction number by estimating the probability of transmission across all infector-infectee pairs (as in infection network) and then the estimation of the relative likelihood of each pair and its summation. Thus, it evades any assumptions of exponentiality which is an advantage over the EG method. The TD method is sometimes rated as the least biased method yet the $R_{(t)}$ calculated by this method seems to be volatile and sensitive as it may change very rapidly even within shorter periods owing to any super-spreading / under-declaring events. These fluctuations in $R_{(t)}$ estimated through TD method may become more evident in this case as the data is crowd-sourced [22].

The results of this study should be interpreted with certain caveats apart from the inherent limitations of crowd-sourced nature of the data. The credibility of a crowd-sourced dataset may be viewed from the following perspectives: under-reporting, duplicated / redundant information, incomplete information, differential lag in reporting the cases, missing initial cases, the inclusion of imported cases as native cases, and partisan information. These may lead to overestimation or underestimation of reproduction numbers. However, as the cases in this particular instance (www.covid19india.org) are chiefly pulled from official government handles, the extent of discrepancy may remain the same in some dimensions irrespective of the nature of the data source. Secondly, the investigators also tried to minimize these discrepancies by rigorous data cleaning, removal of imported cases (as reported) as far as possible by triangulating with other sources and subsequent merging of the final dataset. The estimates might be influenced by certain effect modifiers and confounders like population density, climatic

variations and violation of the assumption of random mixing. Conceptually, this phenomenon is dynamic and non-linear and hence should be read with caution [20, 23]. The estimated transmission parameters (including doubling time during early outbreak period) in some states show a wider confidence interval with higher uncertainty. One of the plausible reason behind this uncertainty may be that initially number of new cases follows the Poisson process where the approximate average time between events is known but the case to case timing varies significantly at the beginning of the epidemic.

The overall picture suggests the initial success of Indian states to curtail the rise of the curve. However, as a whole, the time-varying reproduction numbers (at the time of last access to the database) were above the epidemic potential. Moreover, every estimation like this has an element of innate variation, grounded in epistemic uncertainties and assumptions of the model. With these caveats, this reduction can mainly be explained by the reduced number of contacts among people owing to movement restrictions. Studies on the impact of lockdown in other countries also reported a reduction in reproduction number, which translates into flattening of the curve and delaying of the peak [24–28]. However, as mentioned earlier, the time-varying reproduction number ($R_{(t)}$) estimations are dynamic and may change over age structure, time and nature of the intervention. $R_{(t)}$ is a measure of transmissibility or contagiousness at a given period, and its reduction should be interpreted with caution. This is indicative of the relative force of infection at a given time while the 'absolute' burden of infections also depend on the duration of infectiousness and progression of time from the first reported case by influencing the mixing probability of infected-infectee pair. This mixing probability is further influenced by population density, mobility patterns and the general population's compliance with the non-pharmaceutical interventions (NPIs). When non-pharmaceutical interventions (NPIs) are enforced, there is a reduction in the number of potential contacts and thereby reducing the $R_{(t)}$. However, in a scenario where $R_{(t)} > 1$, and the number of actively infected persons is high, cases will still rise as one person transmits the infection to one more person. Therefore, in the post-lockdown era, it might be a challenge to maintain this path, and this may be the period where the absolute burden of the infected persons will be high [29, 30]. Also, there has been a disproportionately higher burden of serious infections, including those requiring intensive-care among individuals more than 60 years of age as compared to younger adults [31]. This, coupled with the higher prevalence of comorbid conditions (50%) in individuals over 60 years in India, may warrant a strategy tailored to this section of the population [32]. This also suggests that in addition to the identification of infection, it is imperative to shift the focus on mortality prevention. Containment strategies like lockdown have given us the much-needed opportunity to delay the peak and flatten the epidemic-curve. The time bought should be utilized to intensify the surveillance among 'at-risk' individuals and buttress the health infrastructure, including hospital beds with oxygen availability and critical care beds with ventilators and telemedicine [33–35].

At this juncture, an empirical question arises whether (despite showing the initial success) should the stringent lockdown be continued for a more extended period? Considering the undesired collateral effects of stringent restrictions on the economy and livelihoods of the general population; a nationwide lockdown may not be a feasible solution for a longer duration. Other NPIs (social distancing measures, wearing masks, legal enforcement to curtails the non-essential gatherings, etc.) should be enforced to compensate for the increased probability of random mixing. The decision on which NPI measure should be enforced should vary with the burden of active infections, emerging patterns of severity /mortality, and health system endurance and capacity to deal with such cases embedded in socio-economic and socio-cultural vulnerability.

Another relevant observation in Indian COVID-19 context is that it does not look like an outbreak with similar intensity at the pan-country level. It seems to be a complex aggregation

of several individual outbreaks occurring at different time points at different geographic locations. In principle, the magnitude of these outbreaks should be influenced by population density (outbreaks first started in areas where the population density is high), mobility patterns (higher number of cases were seen in places with better connectivity, i.e. international flights and domestic public transport systems) and the response of the healthcare system, all of which vary across different geographic locations. There is an urgent need for a real-time monitoring system that would take into consideration the disease burden (incidence and mortality), transmission parameters (reproduction number, doubling time and growth rate), existing health infrastructure (including bed capacity, human resources, etc.) and the vulnerability of other essential and frontline sectors [36]. This dynamic monitoring environment could serve as a sensitive tool to detect changes in the epidemiological pathways of COVID-19 and therefore, may facilitate the decision-making process on the nature and extent of NPI enforcement. This statement becomes more pertinent with the findings of our study, where we witness varying trajectories across the ten selected Indian states in response to the nationwide lockdown. Thus, logically the NPI enforcement should be tailored and customized according to the transmission parameters of smaller geographical areas, and hence the proposed monitoring system may play a pivotal role in this regard.

## Conclusion

The current study shows that the epidemic curtailment strategies and lockdown enforced by the Indian government have been effective in reducing the explored transmission parameters. However, the $R_{(t)}$ remains to be above 1. There is also a variation in the decrease of these transmission parameters across different Indian states. With the inevitability of ending a nationwide lockdown, the future mitigation measures may consider this information and develop tailored strategies as alert systems for the institution of NPIs at the state level or even the district level.

## Supporting information

**S1 Appendix. R-code used for estimation of epidemiological parameters and generation of composite plot.**
(PDF)

**S2 Appendix. R code for incidence modelling and future projections (10 days).**
(RMD)

**S1 Table. Number of imported and local cases.**
(PDF)

**S2 Table. Number of projected cases and actual cases.**
(PDF)

**S1 Data.**
(ZIP)

## Acknowledgments

Datasets of this study were extracted from API provided by www.covid19india.org/. This site uses count data published in state bulletins and official handles. It has provided an API for public use of the data. Authors would like to extend thanks to owners of the website and also to many unknown volunteers who does work for validation.

## Author Contributions

**Conceptualization:** Arun Mitra, Ankur Joshi.

**Data curation:** Arun Mitra, Abhijit P. Pakhare, Adrija Roy, Ankur Joshi.

**Formal analysis:** Arun Mitra, Adrija Roy, Ankur Joshi.

**Investigation:** Ankur Joshi.

**Methodology:** Abhijit P. Pakhare, Adrija Roy, Ankur Joshi.

**Validation:** Abhijit P. Pakhare, Adrija Roy.

**Visualization:** Arun Mitra, Abhijit P. Pakhare, Ankur Joshi.

**Writing – original draft:** Arun Mitra, Ankur Joshi.

**Writing – review & editing:** Arun Mitra, Abhijit P. Pakhare, Adrija Roy, Ankur Joshi.

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
