## [Decision Letter · Decision Letter 0]

1 Jul 2020

PONE-D-20-14797

Impact of COVID-19 epidemic curtailment strategies in selected Indian states: an analysis by reproduction number and doubling time with incidence modelling

PLOS ONE

Dear Dr. Joshi,

Thank you for submitting your manuscript to PLOS ONE. After careful consideration, we feel that it has merit but does not fully meet PLOS ONE’s publication criteria as it currently stands. Therefore, we invite you to submit a revised version of the manuscript that addresses the points raised during the review process.

We look forward to receiving your revised manuscript.

Kind regards,

Shinya Tsuzuki, MD, MSc

Academic Editor

PLOS ONE

Journal Requirements:

Additional Editor Comments (if provided):

Basically I agree with the points both reviewers pointed out then they should be addressed before publication.

Reviewers' comments:

Reviewer's Responses to Questions

**Comments to the Author**

1. Is the manuscript technically sound, and do the data support the conclusions?

Reviewer #1: Yes

Reviewer #2: Yes

2. Has the statistical analysis been performed appropriately and rigorously? 

Reviewer #1: No

Reviewer #2: Yes

3. Have the authors made all data underlying the findings in their manuscript fully available?

Reviewer #1: Yes

Reviewer #2: No

4. Is the manuscript presented in an intelligible fashion and written in standard English?

Reviewer #1: Yes

Reviewer #2: Yes

5. Review Comments to the Author

Reviewer #1: There are a number of small issues in the writing which should be cleared up before publication, but overall the paper is readable and reasonably constructed.

The bigger issue is that relatively little information is given about the actual methodology, the benefits and drawbacks, alternatives, and general conclusions. Two R packages are used to fit epidemic models to a particular data source, but the methods employed in each are not really described in any detail. Some statements are extremely vague: "This method optimizes β and S0 from the sequence of binomial likelihood with the fundamental assumptions of conditional independence" - that's not enough information to describe the form of the likelihood, or the maximization scheme. The language describing the methods remains confusing into the discussion section ("yet the methods are robust in terms of conditional independence and MCMC methods used to tackle the Bayesian influence").

The data itself is described as "crowd sourced", but this is not sufficiently descriptive. The data appears to be collected from official sources by volunteers - that's fine, but a discussion of the limitations of official figures is certainly important with COVID-19. Testing and ascertainment has changed throughout the course of the pandemic, but the potentially incomplete data are simply treated as known measures of incidence here.

In general, the work does not seem unreasonable, but the methods descriptions are generally lacking and need to be improved and expanded. It's also not clear how impactful the findings as presented are - lock-downs have been widely studied in COVID-19, and the general consensus already shows that they've decreased transmission. More contextualization to the situation in Indian and specific implications of this work would improve the manuscript.

Reviewer #2: The authors used an R package “R0” to estimate the reproduction number of COVID-19 in 10 states in India, 15 days and 30 days into the “lockdown” respectively, through a maximum likelihood method based on chain binomial models.

The authors used an R package “projections” to simulate 1000 probable epidemic outbreak trajectories, based on probability mass function dependent branching process assuming it follows a Poisson distribution. Based on data as of April 23, 2020, the authors projected the cumulative incidence 10 days into the future (May 3, 2020).

The authors demonstrate their command of advanced analytical skills through their analysis of the epidemiological data. However, there is room for improvement with regard to careful and clear explanations of concepts and codes therein.

Major comments:

Page 3: “The serial interval for COVID-19 … was used as time stamp for the estimation.” Could you please explain what do you mean by “time stamp” here?

Can you please clarify if you are using serial interval as a proxy for generation time?

Minor comments:

Page 3: Title of Table 1. Please include: “as of 23rd April 2020” in the title of Table 1. Because a table needs to be standalone and without the date of data it will not be meaningful.

Page 3: “We used the package incidence to model the incidence and estimate growth rate and doubling time and package R0 to estimate the reproduction number (R0) for different states. (7–9) The package projections was used to simulate the epidemic outbreaks and project their respective trajectories based on the state specific transmission parameters.(10)” Here the names of the R package, “incidence”, “R0” and “projections” should be in quotes. Otherwise, it is confusing to the readers.

Page 4: “v-line”: change to “vertical lines”

Page 4: Table 2: Is the range in square brackets, a range or a confidence intervals. Reproduction number for Tamil Nadu: why was it NA but with the range in square brackets?

Page 5: Figure 1. Please provide more details in the legend of the figure. A figure should be standalone. The legend should explain that line graphs represent cumulative cases and bar charts represent new cases per day.

Page 5: “baseline R0 (calculated at 15 days into lockdown) and effective R0 at 30 days into lockdown”: R0 is the symbol of the basic reproduction number, which is defined as the number of secondary cases caused by an index case in a totally susceptible population, without interventions or behavioral change. When India was in 15 days into lockdown or 30 days into lockdown, these assumptions no longer held. Therefore, the reproduction number 15 or 30 days into lockdown should be referred to as an effective reproduction number (Re), or a time-varying or time-dependent reproduction number (Rt). They are not R0 in these cases. Calling them “baseline R0” or “effective R0” are wrong. The comment here also applies to “effective R0” in the Discussion (e.g., Page 7, the 3rd line in the 1st paragraph in the Discussion; also in other places).

Page 7: R_t as the symbol for effective reproduction numbers is introduced. So why would you not use it consistently across the entire paper?

Page 7-8: The end of page 7: “to delay the peak flatten the epidemic-curve”: Please change to “to delay the peak and flatten the epidemic curve”.

The following 2 medRxiv preprint manuscripts will be helpful to you in clarifying different concepts and the statistical methods that estimated them.

Practical considerations for measuring the effective reproductive number, Rt

doi: https://doi.org/10.1101/2020.06.18.20134858

A comparative analysis of statistical methods to estimate the reproduction number in emerging epidemics with implications for the current COVID-19 pandemic

doi: https://doi.org/10.1101/2020.05.13.20101121

Finally, the authors may want to reconsider the flow of their Discussion section. I would recommend the following flow:

Discussion paragraph #1: Highlights of the key results of the paper

Discussion paragraph #2: Situating your manuscript in the body of recent literature of COVID-19 epidemiology, esp. other papers that estimate the R_t of COVID-19

Discussion paragraph #3: Limitations of the paper

Discussion paragraph #4: Conclusions

Supplementary Appendix 1

Some of the codes go beyond the margin and there are incomplete in the PDF file (these happen multiple times in multiple pages). Please amend.

Section heading “Creating incidence objectes…”: “objectes” should be “objects”.

Section “Serial Interval”: “The serial interval is ceated using…”: “ceated” should be “created”.

Can we say that you are using the serial interval to approximate the generation time?

Page 6:

# Aggregate the timepoints after enforcment of lockdown

plot_states_14 <- readRDS("transmission_params_14.rds")

plot_states_30 <- readRDS("transmission_params_30.rds")

Where does the “transmission_params_14.rds” and “transmission_params_30.rds” files come from? I cannot find it in the previous code chunks.

Please make sure that you introduce them in the text and explain what they are.

Supplementary Appendix 2

“Under Loading Packages & .rds files” What are the RDS files here? You need to introduce to your readers what they are: “df_backup.rds”, “transmission_params_30.rds”, and “states_keep.rds”.

To comply with PLOS ONE requirement to make the data underlying the findings in the manuscript fully available, have you considered also provide these RDS files as supplementary materials? Otherwise, even with your R codes, your analysis cannot be repeated by the readers.

6. PLOS authors have the option to publish the peer review history of their article (what does this mean?). If published, this will include your full peer review and any attached files.

Reviewer #1: No

Reviewer #2: **Yes: **Isaac Chun-Hai Fung

---

## [Author Response · Author response to Decision Letter 0]

17 Jul 2020

Response to Reviewers

At the outset we would like to thank the editor and the reviewers for their valuable comments, it is indeed an honour to receive critique and suggestions from esteemed scholars like you. We are also grateful for giving us the motivation continue playing our little part in fighting the COVID-19 epidemic. 

Editor Comments:

Response:

The authors apologize for the errors in the submission of the manuscript. We have formatted and resubmitted the manuscript according to the PLOS ONE style template as suggested.

Additional Editor Comments:

Basically I agree with the points both reviewers pointed out then they should be addressed before publication.

Response: 

We attempted to address all the issues raised by the editor and reviewers to the best of our abilities. We have performed a thorough literature review and rewritten the methods and discussion part, incorporating all the suggestions. We also described the R code and the accompanying files in greater detail, updated the Supplementary Appendix. We published the steps employed in the computational workflow of the analysis at protocols.io and uploaded data and other relevant files adhering to the PLOS One’s publication policy.

Reviewer comments:

Reviewer #1: 

There are a number of small issues in the writing which should be cleared up before publication, but overall the paper is readable and reasonably constructed.

Response:

We thank the Reviewer for their encouraging words. We have taken all your suggestions and rectified the manuscript. We hope we addressed the issues pointed adequately. 

Point 1:

The bigger issue is that relatively little information is given about the actual methodology, the benefits and drawbacks, alternatives, and general conclusions. Two R packages are used to fit epidemic models to a particular data source, but the methods employed in each are not really described in any detail. Some statements are extremely vague: "This method optimizes β and S0 from the sequence of binomial likelihood with the fundamental assumptions of conditional independence" - that's not enough information to describe the form of the likelihood, or the maximization scheme. The language describing the methods remains confusing into the discussion section ("yet the methods are robust in terms of conditional independence and MCMC methods used to tackle the Bayesian influence").

Response:

The authors thank the Reviewer for his/her suggestions. We have rewritten the Methodology section in a more lucid language and also tried to explain the rationale behind our methodology. We also elaborated on the considerations and assumptions in greater detail and included the detailed descriptions of the elements of the R code in the Supplementary Appendix. 

The relevant part of the Methdology Section is given below:

Estimation of Reproduction Number 

The time-varying reproductive number (R(t)) was estimated by using the maximum likelihood (ML) method. [13] This method presumes that all the secondary cases linked to the primary cases follow a Poisson process (event rate is constant) and the corresponding serial interval follows a multinomial distribution. This leads to a gradual trend towards zero secondary cases (with multitude of time steps) arising from primary cases during a specific time-step. The gradient of depends on the parameter distribution function (PDF) of the serial interval. The “est R0.ML()” function in the “R0” package. This runs an expectation maximum (EM) algorithm which maximises the distribution probability of primary and secondary cases in reference to time. This method assumes that infectee always develops symptoms only after infector thus the value of the serial interval will be positive. 

Serial Interval 

For the parameter distribution function (PDF), we could not obtain the generation time (time lag between the infection in primary case and secondary cases) distribution directly with infector-infectee pairs due to the lack of data availability. Therefore, it was substituted with the serial interval distribution discretized on a 1-day time step. This was created using the generation.time()” function in the “R0” package. For parametrisation purposes we chose a gamma distribution as it accommodates for the underlying changing number of events in the constant event rate (Poisson process). The distribution assumptions were aligned with the emerging literature as well as the observed plausible transmission dynamics. The mean and standard deviation for serial interval approximations were 4.4 days and 3 days respectively.[14] The shape (number of events in time step) and scale (the reciprocal of event rate) of the distribution were 2.15 and 2.04 respectively. 

Point 2:

The data itself is described as "crowd sourced", but this is not sufficiently descriptive. The data appears to be collected from official sources by volunteers - that's fine, but a discussion of the limitations of official figures is certainly important with COVID-19. Testing and ascertainment has changed throughout the course of the pandemic, but the potentially incomplete data are simply treated as known measures of incidence here.

Response:

The authors agree with the Reviewer’s comment regarding the insufficient description of the data source. 

We have added the following text in the Methdology section and hope it describes the data source sufficiently and addresses the concerns raised:

Data Source

The data source is a crowd-sourced database maintained at www.covid19india.org. [6] The website is maintained by a group of volunteers who curate and verify the data coming from several sources. The data is compiled from the state bulletins, official handles of state governments, and health ministries. It is validated, updated periodically and published into an application programming interface (API) and Google spreadsheet which is accessible at api.covid19india.org for the public. Apart from the patient-level data, the API includes district-level, state-level, and national-level datasets. We used the data from the line-listing of the cases reported as positive for COVID-19. The data was iteratively and progressively accessed through the database in coherence with creation and improvement in analysis code. The last access to database was made on 1st May, 2020. We truncated the data up to 23th April 2020 for the purpose of this study. This buffer period of 7 days offered some immunity against the possible delay to add the cases and our limitation to access the data in real time.

We also attempted to address the implications of testing, quality of the data on the estimation of these vital parameters in the Discussion section. The relevant portion of the manuscript is given below:

The credibility of a crowd-sourced dataset may be viewed from the following probes: under-reporting, duplicated / redundant information, incomplete information, differential lag in reporting the cases, missing initial cases, the inclusion of imported cases as native cases, and partisan information. These all probes may lead to overestimation or underestimation of reproductive numbers. However, as the cases in this particular instance (www.covid19india.org) are chiefly pulled from official government handles, the extent of discrepancy may remain the same in some probes irrespective of the nature of the data source. Secondly, the investigators also tried to minimize these discrepancies by rigorous data cleaning, removal of imported cases (as reported) as far as possible by triangulating with other sources and subsequent merging of the final dataset. The estimates might be influenced by certain effect modifiers and confounders like population density, climatic variations and violation of assumption of random mixing. Conceptually, this phenomenon is dynamic and non-linear in nature and hence should be read with caution. 

Point 3:

In general, the work does not seem unreasonable, but the methods descriptions are generally lacking and need to be improved and expanded. It's also not clear how impactful the findings as presented are - lock-downs have been widely studied in COVID-19, and the general consensus already shows that they've decreased transmission. More contextualization to the situation in Indian and specific implications of this work would improve the manuscript.

Response:

We thank the Reviewer for his/her encouraging words and suggestions for improving our manuscript. We have redrafted the Discussion section adding more context into the situation in India and the implications of our work. The relevant parts of the Discussion are provided below:

The overall picture suggests the initial success of Indian states to curtail the rise of curve. This reduction can mainly be explained by reduced number of contacts among people owing to movement restrictions. Studies on the impact of lockdown in other countries also reported reduction in reproductive number which translates in to flattening of the curve and delaying of peak.[20-24] Yet as mentioned earlier, the time-varying reproduction number (R(t)) estimations are dynamic and may change over age structure, time and nature of intervention. Continuing nationwide lockdown will not be feasible in long term and restrictions have to be eased in phase wise manner. Therefore, in the post-lockdown era, it might be a challenge to maintain this path and this may be the period where the absolute burden of the infected persons will be high.[25, 26] Also, there has been disproportionately higher burden of serious infections including those requiring intensive-care among individuals more than 60 years of age as compared to younger adults.[27] This coupled with the higher prevalence of comorbid conditions (50%) in individuals over 60 years in India may warrant a strategy tailored to this section of population. [28] This also suggests that in addition to identification of infection, it is imperative to shift the focus on mortality prevention. Containment strategies like lockdown has given us the much-needed opportunity to delay the peak flatten the epidemic-curve. The time bought should be utilized to intensify the surveillance among `at-risk' individuals and buttress the health infrastructure including hospital beds with oxygen availability and critical care beds with ventilators and tele-medicine. [29-31]

 

Reviewer #2: 

The authors used an R package “R0” to estimate the reproduction number of COVID-19 in 10 states in India, 15 days and 30 days into the “lockdown” respectively, through a maximum likelihood method based on chain binomial models. 

The authors used an R package “projections” to simulate 1000 probable epidemic outbreak trajectories, based on probability mass function dependent branching process assuming it follows a Poisson distribution. Based on data as of April 23, 2020, the authors projected the cumulative incidence 10 days into the future (May 3, 2020).

The authors demonstrate their command of advanced analytical skills through their analysis of the epidemiological data. However, there is room for improvement with regard to careful and clear explanations of concepts and codes therein.

Response:

The authors would like to thank the Reviewer for his/her comments and suggestions in making our manuscript better. We tried to address all the issues raised and hope our responses are adequate.

Point 1:

Page 3: “The serial interval for COVID-19 … was used as time stamp for the estimation.” Could you please explain what do you mean by “time stamp” here?

Can you please clarify if you are using serial interval as a proxy for generation time?

Response: 

By “timestamp” we meant “time-step”. The authors apologize for the error in the text. We have used the serial interval as a proxy for generation time. We thank the Reviewer for pointing it out. We have rewritten the relevant part of the methodology as below:

Serial Interval 

For the parameter distribution function (PDF), we could not obtain the generation time (time lag between the infection in primary case and secondary cases) distribution directly with infector-infectee pairs due to the lack of data availability. Therefore, it was substituted with the serial interval distribution discretized on a 1-day time step. This was created using the generation.time()” function in the “R0” package. For parametrisation purposes we chose a gamma distribution as it accommodates for the underlying changing number of events in the constant event rate (Poisson process). The distribution assumptions were aligned with the emerging literature as well as the observed plausible transmission dynamics. The mean and standard deviation for serial interval approximations were 4.4 days and 3 days respectively.[14] The shape (number of events in time step) and scale (the reciprocal of event rate) of the distribution were 2.15 and 2.04 respectively. 

Point 2:

Page 3: Title of Table 1. Please include: “as of 23rd April 2020” in the title of Table 1. Because a table needs to be standalone and without the date of data it will not be meaningful.

Response:

The authors apologize for the typographical error and rectified it in the resubmitted Manuscript.

Point 3:

Page 3: “We used the package incidence to model the incidence and estimate growth rate and doubling time and package R0 to estimate the reproduction number (R0) for different states. (7–9) The package projections was used to simulate the epidemic outbreaks and project their respective trajectories based on the state specific transmission parameters.(10)” Here the names of the R package, “incidence”, “R0” and “projections” should be in quotes. Otherwise, it is confusing to the readers.

Response:

The authors apologize for the typographical error and rectified it in the resubmitted Manuscript.

Point 4:

Page 4: “v-line”: change to “vertical lines”

Response:

The authors apologize for the typographical error and rectified it in the resubmitted Manuscript.

Point 5:

Page 4: Table 2: Is the range in square brackets, a range or a confidence intervals. Reproduction number for Tamil Nadu: why was it NA but with the range in square brackets?

Response:

Table has been rectified now.

Point 6:

Page 5: Figure 1. Please provide more details in the legend of the figure. A figure should be standalone. The legend should explain that line graphs represent cumulative cases and bar charts represent new cases per day.

Response:

The authors for the typographical error and rectified it in the resubmitted Manuscript. We have added the legend separately for both daily incidence and cumulative incidence. The revised Figure is given below: 

Point 7:

Page 5: “baseline R0 (calculated at 15 days into lockdown) and effective R0 at 30 days into lockdown”: R0 is the symbol of the basic reproduction number, which is defined as the number of secondary cases caused by an index case in a totally susceptible population, without interventions or behavioral change. When India was in 15 days into lockdown or 30 days into lockdown, these assumptions no longer held. Therefore, the reproduction number 15 or 30 days into lockdown should be referred to as an effective reproduction number (Re), or a time-varying or time-dependent reproduction number (Rt). They are not R0 in these cases. Calling them “baseline R0” or “effective R0” are wrong. The comment here also applies to “effective R0” in the Discussion (e.g., Page 7, the 3rd line in the 1st paragraph in the Discussion; also in other places).

Response:

Now we have used term R(t) (time-varying reproduction number) for 15 and 30 days into lockdown. Changes have been made throughout the manuscript.

Point 8:

Page 7: R_t as the symbol for effective reproduction numbers is introduced. So why would you not use it consistently across the entire paper?

Response: Now, we have used term R(t) for effective reproduction number throughout the manuscript. 

Point 9:

Page 7-8: The end of page 7: “to delay the peak flatten the epidemic-curve”: Please change to “to delay the peak and flatten the epidemic curve”.

Response:

The authors for the typographical error and rectified it in the resubmitted Manuscript.

Point 10:

The following 2 medRxiv preprint manuscripts will be helpful to you in clarifying different concepts and the statistical methods that estimated them.

Practical considerations for measuring the effective reproductive number, Rt

doi: https://doi.org/10.1101/2020.06.18.20134858

A comparative analysis of statistical methods to estimate the reproduction number in emerging epidemics with implications for the current COVID-19 pandemic

doi: https://doi.org/10.1101/2020.05.13.20101121

Response:

We are greatful to reviwers for suggesting these papers. These helped in revising the manuscript particularly with reference explanation of methods for estimation of reproduction numbers. We have cited these at appropriate sections in the manuscript.

Point 11:

Finally, the authors may want to reconsider the flow of their Discussion section. I would recommend the following flow:

Discussion paragraph #1: Highlights of the key results of the paper

Discussion paragraph #2: Situating your manuscript in the body of recent literature of COVID-19 epidemiology, esp. other papers that estimate the R_t of COVID-19

Discussion paragraph #3: Limitations of the paper

Discussion paragraph #4: Conclusions

Response:

The authors thank the Reviewer for his/her suggestions. We have taken your advice and redrafted the Discussion section. We included recent papers on Indian context estimating time-varying reproduction number (R(t)). We also addressed the issue of crowd-sourced data in the limitations section of the Discussion. We hope the corrections made are adequate and address the issues raised. 

Point 12:

Supplementary Appendix 1

Some of the codes go beyond the margin and there are incomplete in the PDF file (these happen multiple times in multiple pages). Please amend.

Section heading “Creating incidence objectes…”: “objectes” should be “objects”.

Section “Serial Interval”: “The serial interval is ceated using…”: “ceated” should be “created”.

Can we say that you are using the serial interval to approximate the generation time?

Response:

We thank the Reviewer for his/her comments. We apologize for the typographical errors. We ensured that all the code stays within the margins of the PDF document. Regarding the Serial Interval being used as an approximation of Generation Time, we have mentioned the same in the Methodology section as suggested by the Reviewer. We have made the necessary corrections in the resubmitted manuscript and Supplementary Appendix.

Point 13:

Page 6:

# Aggregate the timepoints after enforcment of lockdown

plot_states_14 <- readRDS("transmission_params_14.rds")

plot_states_30 <- readRDS("transmission_params_30.rds")

Where does the “transmission_params_14.rds” and “transmission_params_30.rds” files come from? I cannot find it in the previous code chunks.

Please make sure that you introduce them in the text and explain what they are.

Response:

The “transmission_params_14.rds” and “transmission_params_30.rds” are summary tables of transmission parameters of 15 days and 30 days into the epidemic. These objects were created in the R code provided in Supplementary Appendix 1. We apologize for not considering the issue raised by the Reviewer earlier. We realise our mistake and amended it. We have provided the concerned *.rds files, their description and the corresponding code at the data repository at protocols.io where we provided the description and computational workflow along with the necessary files (https://protocols.io/view/incidence-modelling-covid19-computational-workflow-bh8vj9w6). We also refined the R code chunks and explained them in greater detail in the updated Supplementary Appendix.

Point 14:

Supplementary Appendix 2

“Under Loading Packages & .rds files” What are the RDS files here? You need to introduce to your readers what they are: “df_backup.rds”, “transmission_params_30.rds”, and “states_keep.rds”.

To comply with PLOS ONE requirement to make the data underlying the findings in the manuscript fully available, have you considered also provide these RDS files as supplementary materials? Otherwise, even with your R codes, your analysis cannot be repeated by the readers.

Response:

The authors apologize for overlooking the issue raised. We have taken your suggestions and elaborated on the RDS files and the corresponding R code. We have also submitted all relevant files necessary for reproducibility at protocols.io (https://protocols.io/view/incidence-modelling-covid19-computational-workflow-bh8vj9w6)

We have added the following text in the Supplementary Appendix 2:

The df_new_05May.rds file contains line listing data till 5th May 2020. The variables are same as described in Supplementary Appendix 1. transmission_params_30.rds file contains the transmission parameters (time-varying reproduction number, doubling time, and growth rate) of the ten selected Indian states estimated after 30 days of lockdown. The states_keep.rds file contains the list of selected Indian states as mentioned in the manuscript and Supplementary Appendix 1.

---

## [Decision Letter · Decision Letter 1]

11 Aug 2020

PONE-D-20-14797R1

Impact of COVID-19 epidemic curtailment strategies in selected Indian states: an analysis by reproduction number and doubling time with incidence modelling

PLOS ONE

Dear Dr. Joshi,

Thank you for submitting your manuscript to PLOS ONE. After careful consideration, we feel that it has merit but does not fully meet PLOS ONE’s publication criteria as it currently stands. Therefore, we invite you to submit a revised version of the manuscript that addresses the points raised during the review process.

Both reviewer raised some minor concerns should be addressed before publication and I agree with their opinions then further minor revision would be desirable.

We look forward to receiving your revised manuscript.

Kind regards,

Shinya Tsuzuki, MD, MSc

Academic Editor

PLOS ONE

Reviewers' comments:

Reviewer's Responses to Questions

**Comments to the Author**

1. If the authors have adequately addressed your comments raised in a previous round of review and you feel that this manuscript is now acceptable for publication, you may indicate that here to bypass the “Comments to the Author” section, enter your conflict of interest statement in the “Confidential to Editor” section, and submit your "Accept" recommendation.

Reviewer #1: (No Response)

Reviewer #3: All comments have been addressed

2. Is the manuscript technically sound, and do the data support the conclusions?

Reviewer #1: Yes

Reviewer #3: Partly

3. Has the statistical analysis been performed appropriately and rigorously? 

Reviewer #1: Yes

Reviewer #3: Yes

4. Have the authors made all data underlying the findings in their manuscript fully available?

Reviewer #1: Yes

Reviewer #3: Yes

5. Is the manuscript presented in an intelligible fashion and written in standard English?

Reviewer #1: Yes

Reviewer #3: No

6. Review Comments to the Author

Reviewer #1: In general, the authors have greatly improved this manuscript. There are a few locations where the language needs to be presented more cautiously, and a few issues with word usage. I greatly appreciated the improved discussion and conclusion section.

1. Absract - While I appreciate the improved language around R(t), it leads to some issues. In the abstract, the authors state: "The chosen transmission

parameters were: time-varying reproduction number (R(t)), doubling time and growth rate." R(t) isn't a single parameter, it's a vector which is estimated. I believe the authors, through the software employed, did indeed parameterize the epidemic through R0, doubling time, and growth rate. I also think this description is still too vague, since it doesn't tell the reader what kind of model (likelihood) is actually being used. This is clarified later in the manuscript, but could be mentioned here.

2. Abstract - "The [estimated] time-varying reproductive numbers are still above the threshold of 1" - this kind of language should be included everywhere such estimates are presented. All estimates are subject to uncertainty, and are based on particular model assumptions.

3. Data Analysis - include parenthetical citation for R (see: 'citation()')

4. General comment - PDF does not generally stand for "parameter distribution function" anywhere that I'm aware of. It generally means "probability density function".

5. Discussion - the new language about the data sources uses the word "probes" incorrectly. Words like "perspectives", or "dimensions" would be more appropriate.

Reviewer #3: Thank you for addressing the reviewer comments and for the draft. While you have responded to the previous queries and suggestions, a few minor comments still remain:

1. In the beginning of the results section, you state "23,040 COVID-19 cases have been reported in India as of 23rd April 2020 of which 20,590 cases (89.4%) were seen in the selected 10 states". It is still unclear though from the methods section how the data is reported at the state level and whether it is subject to reporting bias by state - therefore, it doesn't seem like the relevant comparison is the number of cases in the 10 highest incident states compared to the rest of India. Please clarify still in the methods section how the case data is collected and reported in India at the state level and what biases exist. Are these cases that get tested and reported at hospitals or local healthcare facilities? Therefore, is it safe to assume that limited healthcare access to such facilities would result in a gross underestimation of cases? Please expand on this in your discussion.

2. Furthermore, please detail even more in the data section of the methods what time to event data was included in the data source (i.e. symptom onset, testing date, generation time, etc) to help explain that you chose distributions for serial interval or generation time not based on data. It appears that case data was used to model and forecast incidence through data fitting and to estimate Rt. This should be clear in the methods what the data was used for.

3. For Figure 1, could you make the state colors in the bar plot showing incidence cases match the line color for each state showing cumulative cases.

4. You show estimates for doubling time in the results, but still only mention the package/function used and not how it was estimated in the methods section.

5. The "Epidemiologic Parameters" section of your results could benefit from reorganization. Try discussing the doubling time results all together. Furthermore, the following statement "This may be due to events following the Poisson process at the beginning of the epidemic where the approximate average time between events is known but the case to case timing varies significantly" belongs in the discussion.

6. In the discussion, you don't quite address why the the lockdown resulted in a reduction in Rt but a continual rise in cases. Please expand on this. Given the objective of your paper, it also seems important to explicitly state that implementing a lockdown strategy at the beginning of the epidemic was critical but in the longterm for economic reasons may not be feasible, particularly given the density of the population and economic disparities in India. You make the following statement "Continuing a nationwide lockdown would not be feasible in the long term, and restrictions have to be eased in a phase-wise manner," but provide no references or further explanation for why it is not feasible and why restrictions should be phase-wise or examples. One of the most interesting/important aspects of understanding transmission in India is the population density and mobility patterns of the population - it would be helpful to expand on how your results apply to this or how future surveillance and studies should account for this. Why is knowing your results useful for future non-pharmaceutical interventions (NPIs) in India particularly when lockdown has historically shown the greatest reduction for this epidemic and the 1918 Flu and you're suggesting it's not feasible in India to maintain? Please edit your discussion/conclusion to address this.

7. PLOS authors have the option to publish the peer review history of their article (what does this mean?). If published, this will include your full peer review and any attached files.

Reviewer #1: No

Reviewer #3: No

---

## [Author Response · Author response to Decision Letter 1]

21 Aug 2020

Reviewer #1

In general, the authors have greatly improved this manuscript. There are a few locations where the language needs to be presented more cautiously, and a few issues with word usage. I greatly appreciated the improved discussion and conclusion section.

Response:

We thank the reviewer for his/her guidance in improving this manuscript. Your encouraging words give us great motivation in our work. We tried to address all the concerns raised and updated the abstract. We hope the new additions provide better clarity to the reader.

Point 1:

Absract - While I appreciate the improved language around R(t), it leads to some issues. In the abstract, the authors state: "The chosen transmission parameters were: time-varying reproduction number (R(t)), doubling time and growth rate." R(t) isn't a single parameter, it's a vector which is estimated. I believe the authors, through the software employed, did indeed parameterize the epidemic through R0, doubling time, and growth rate. I also think this description is still too vague, since it doesn't tell the reader what kind of model (likelihood) is actually being used. This is clarified later in the manuscript, but could be mentioned here.

Response:

We thank the reviewer for his/her suggestions. We updated the abstract by adding the highlighted text relating to the issue raised in-order to provide better clarity.

Abstract

The Government of India in-network with the state governments has implemented the epidemic curtailment strategies inclusive of case-isolation, quarantine and lockdown in response to ongoing novel coronavirus (COVID-19) outbreak. In this manuscript, we attempt to estimate the impact of these steps across ten selected Indian states using crowd-sourced data. The trajectory of the outbreak was parameterized by the reproduction number (R0), doubling time, and growth rate. These parameters were estimated at two time-periods after the enforcement of the lockdown on 24th March 2020, i.e. 15 days into lockdown and 30 days into lockdown. The authors used a crowd sourced database which is available in the public domain. After preparing the data for analysis, R0 was estimated using maximum likelihood (ML) method which is based on the expectation minimum algorithm where the distribution probability of secondary cases is maximized using the serial interval discretization. The doubling time and growth rate were estimated by the natural log transformation of the exponential growth equation. The overall analysis shows decreasing trends in time-varying reproduction numbers (R(t)) and growth rate (with a few exceptions) and increasing trends in doubling time. The curtailment strategies employed by the Indian government seem to be effective in reducing the transmission parameters of the COVID-19 epidemic. The estimated R(t) are still above the threshold of 1, and the resultant absolute case numbers show an increase with time. Future curtailment and mitigation strategies thus may take into account these findings while formulating further course of action.

Point 2:

Abstract - "The [estimated] time-varying reproductive numbers are still above the threshold of 1" - this kind of language should be included everywhere such estimates are presented. All estimates are subject to uncertainty, and are based on particular model assumptions.

Response:

The authors thank the reviewer for pointing this out. We have addressed the issue and added the following text to the Discussion section:

The estimated transmission parameters (including doubling time during early outbreak period) in some states show a wider confidence interval with higher uncertainty. One of the plausible reason behind this uncertainty may be that initially number of new cases follows the Poisson process where the approximate average time between events is known but the case to case timing varies significantly at the beginning of the epidemic.

The overall picture suggests the initial success of Indian states to curtail the rise of the curve. However, as a whole, the time-varying reproduction numbers (at the time of last access to the database) were above the epidemic potential. Moreover, every estimation like this has an element of innate variation, grounded in epistemic uncertainties and assumptions of the model. With these caveats, this reduction can mainly be explained by the reduced number of contacts among people owing to movement restrictions.

Point 3:

Data Analysis - include parenthetical citation for R (see: 'citation()')

Response:

We thank the reviewer for his/her suggestion. We have added the citation for R at the end of the sentence (reference number 7) in the original manuscript. However, we agree with the reviewer and have included the citation at the appropriate place in the sentence. 

Point 4:

General comment - PDF does not generally stand for "parameter distribution function" anywhere that I'm aware of. It generally means "probability density function".

Response:

We thank the reviewer for pointing out the error, we apologise for the mistake in the abbreviation. We made the necessary correction in the revised manuscript. 

Point 5:

Discussion - the new language about the data sources uses the word "probes" incorrectly. Words like "perspectives", or "dimensions" would be more appropriate.

Response:

We agree with the reviewer regarding the improper use of the word ‘probes’ in the data source section. We agree with the reviewer that the words ‘perspective’ and ‘dimension’ would more appropriate in conveying the message in the sentence. We have taken the suggestion and revised the relevant section as follows: 

The credibility of a crowd-sourced dataset may be viewed from the following perspectives: under-reporting, duplicated / redundant information, incomplete information, differential lag in reporting the cases, missing initial cases, the inclusion of imported cases as native cases, and partisan information. These may lead to overestimation or underestimation of reproduction numbers. However, as the cases in this particular instance (www.covid19india.org) are chiefly pulled from official government handles, the extent of discrepancy may remain the same in some dimensions irrespective of the nature of the data source.

Reviewer #3

Thank you for addressing the reviewer comments and for the draft. While you have responded to the previous queries and suggestions, a few minor comments still remain:

Response:

The authors are grateful for all your suggestions and comments which helped in bettering the manuscript. We thank the reviewer for making it possible. 

Point 1a:

In the beginning of the results section, you state "23,040 COVID-19 cases have been reported in India as of 23rd April 2020 of which 20,590 cases (89.4%) were seen in the selected 10 states". It is still unclear though from the methods section how the data is reported at the state level and whether it is subject to reporting bias by state - therefore, it doesn't seem like the relevant comparison is the number of cases in the 10 highest incident states compared to the rest of India. Please clarify still in the methods section how the case data is collected and reported in India at the state level and what biases exist. 

Response:

The authors thank the reviewer for making this suggestion. We agree with the comment and addressed it by inserting a paragraph on the flow of reporting at the beginning of Data Source section of the Methodology. The relevant text added in the manuscript is provided below:

Methodology Section:

Data Source

COVID-19 cases, deaths and recoveries in India are reported by state public health agencies to the Ministry of Health and Family Welfare (MoHFW), Government of India. The MoHFW releases the testing guidelines and amends them as per the epidemiological scenarios and expert opinion. The eligibility for testing includes patients presenting with suspected symptoms in hospitals, exposed healthcare workers as well as contacts identified through contact tracing. All the hospitals or outreach services notifies cases to district level public health authority which in turn compiles data and reports to state level public health authority. Daily report on number of cases, recoveries and deaths along with the line-list of cases are sent from states to MoHFW on a dedicated portal. State public health authorities simultaneously publishes daily bulletin of same reports.

The data source used for this study is compiled from these state bulletins, official handles of state governments, and health ministries and maintained at www.covid19india.org. [6] This crowd-sourced database and website is maintained by a group of volunteers who curate and verify the data coming from several sources mentioned above.

Point 1b:

Are these cases that get tested and reported at hospitals or local healthcare facilities? Therefore, is it safe to assume that limited healthcare access to such facilities would result in a gross underestimation of cases? Please expand on this in your discussion.

Response:

The reported cases included, cases tested in hospital as well as those tested consequent to contact tracing. One of the significant bias arising due to the usage of reported cases is the delay in reporting between symptom onset, testing and the final diagnosis. The present manuscript pertains to the early outbreak period and therefore we assume that although likelihood of delays in reporting is present, its effect and quantum would be much less as compared to the peak. We have addressed this issue in limitations part of the discussion as-well. 

The relevant text added in the manuscript is provided below:

Discussion Section:

The results of this study should be interpreted with certain caveats apart from the inherent limitations of crowd-sourced nature of the data. The credibility of a crowd-sourced dataset may be viewed from the following perspectives: under-reporting, duplicated / redundant information, incomplete information, differential lag in reporting the cases, missing initial cases, the inclusion of imported cases as native cases, and partisan information. These may lead to overestimation or underestimation of reproduction numbers. However, as the cases in this particular instance (www.covid19india.org) are chiefly pulled from official government handles, the extent of discrepancy may remain the same in some dimensions irrespective of the nature of the data source. Secondly, the investigators also tried to minimize these discrepancies by rigorous data cleaning, removal of imported cases (as reported) as far as possible by triangulating with other sources and subsequent merging of the final dataset. The estimates might be influenced by certain effect modifiers and confounders like population density, climatic variations and violation of the assumption of random mixing. Conceptually, this phenomenon is dynamic and non-linear and hence should be read with caution.[20,23] The estimated transmission parameters (including doubling time during early outbreak period) in some states show a wider confidence interval with higher uncertainty. One of the plausible reason behind this uncertainty may be that initially number of new cases follows the Poisson process where the approximate average time between events is known but the case to case timing varies significantly at the beginning of the epidemic.

Point 2:

Furthermore, please detail even more in the data section of the methods what time to event data was included in the data source (i.e. symptom onset, testing date, generation time, etc) to help explain that you chose distributions for serial interval or generation time not based on data. It appears that case data was used to model and forecast incidence through data fitting and to estimate Rt. This should be clear in the methods what the data was used for.

Response:

The line-list in the database used by us didn’t have complete information on time of symptoms onset or laboratory testing date. We had to rely on the estimates of the serial interval reported in the literature as an approximation for the generation time. However, the database had information relating to the travel history of the individual. Based on this information, we labelled as case ‘imported’ (H/O of travel from COVID-19 affected country) or local. Thus, we created daily incidence objects for each state by selecting only ‘local’ cases for further analysis.

We updated the Methodology to make this clearer.

Data Preparation

The data were prepared for analysis in the following steps:

1. Loading the *.json file containing the raw line-list data 

2. A data-frame is then created and variables of interest are selected 

3. The imported cases are then coded in the following fashion: 

a. All cases with travel history outside the country before the lockdown are coded as imported cases. These cases were removed from further analysis. 

b. All cases reported after 15 days of the lockdown (i.e. 9th April 2020) irrespective of their travel history are coded as local cases 

4. Data from the top 10 states with highest number of cases were subset. 

Respective incidence objects were created by adding number of local cases reported on each date based on the timeframes described below.

Serial Interval

… we could not obtain the generation time (time lag between the infection in the primary case and secondary cases) distribution directly with infector-infectee pairs due to the lack of data availability. Therefore, it was substituted with the serial interval distribution discretized on a 1-day time-step. This was created using the generation.time()" function in the "R0" package. For parametrization purposes, we chose a gamma distribution as it accommodates for the underlying changing number of events in the constant event rate (Poisson process). The distribution assumptions were aligned with the emerging literature as well as the observed plausible transmission dynamics. The mean and standard deviation for serial interval approximations was 4.4 days and 3 days, respectively.[14]

Point 3:

For Figure 1, could you make the state colors in the bar plot showing incidence cases match the line color for each state showing cumulative cases.

Response:

We thank the reviewer for the suggestion. We have matched the colours of the daily incidence (bars) and cumulative incidence (lines) for each state and created a common legend at the bottom of the plot. The newly rendered plot is provided below:

Point 4:

You show estimates for doubling time in the results, but still only mention the package/function used and not how it was estimated in the methods section.

Response:

The authors agree with the comment made by the reviewer. We thank you for pointing out this lack of clarity regarding the method of estimation of doubling time. The fit() function of the incidence package fits an exponential model to the incidence data in the form of: log(y) = r * t + b 

Where; y is the incidence, t is the time (in days) and r is the growth rate while b is the intercept or origin. The doubling time is then estimated by dividing the natural logarithm of 2 with the growth rate of the epidemic i.e. doubling time (d) = log(2)/r. 

This explanation of the method of estimation of doubling time is also added in the methods section as suggested by the reviewers. The relevant section of the methods section is given below:

The growth rate and doubling time were estimated using the "fit()” function of the "incidence” package fits an exponential model to the incidence data in the form of: log(y) = r * t + b ; where y is the incidence, t is the time (in days) and r is the growth rate while b is the intercept or origin. The doubling time is then estimated by dividing the natural logarithm of 2 with the growth rate of the epidemic i.e. doubling time (d) = log(2) / r.

Point 5a:

The "Epidemiologic Parameters" section of your results could benefit from reorganization. Try discussing the doubling time results all together. Furthermore, the following statement "This may be due to events following the Poisson process at the beginning of the epidemic where the approximate average time between events is known but the case to case timing varies significantly" belongs in the discussion.

Response:

We have updated the Epidemiological Parameters section incorporating your suggestions. We added the following text regarding the doubling time in the Epidemiological Parameters section:

Doubling time also changed with the evolving outbreak. Increase in doubling time means a slow growth rate of an outbreak. Five states reported an increase in doubling time, and four states reported negligible change in doubling time. The state of Gujarat reported a decrease in doubling time which could mean that there is no slowdown of the outbreak.

Point 5b:

Furthermore, the following statement "This may be due to events following the Poisson process at the beginning of the epidemic where the approximate average time between events is known but the case to case timing varies significantly" belongs in the discussion.

Response:

We thank the reviewer for making this suggestion. We agree with you and added the following text in the Discussion section:

The estimated transmission parameters (including doubling time during early outbreak period) in some states show a wider confidence interval with higher uncertainty. One of the plausible reason behind this uncertainty may be that initially number of new cases follows the Poisson process where the approximate average time between events is known but the case to case timing varies significantly at the beginning of the epidemic.

Point 6a:

In the discussion, you don't quite address why the the lockdown resulted in a reduction in Rt but a continual rise in cases. Please expand on this

Response:

We have modified discussion section to explain, why cases keep rising despite reduction in R(t) value. The relevant text has been provided below:

R(t) is a measure of transmissibility or contagiousness at a given period, and its reduction should be interpreted with caution. This is indicative of the relative force of infection at a given time while the ‘absolute’ burden of infections also depend on the duration of infectiousness and progression of time from the first reported case by influencing the mixing probability of infected-infectee pair. This mixing probability is further influenced by population density, mobility patterns and the general population’s compliance with the non-pharmaceutical interventions (NPIs). When non-pharmaceutical interventions (NPIs) are enforced, there is a reduction in the number of potential contacts and thereby reducing the R(t). However, in a scenario where R(t) > 1, and the number of actively infected persons is high, cases will still rise as one person transmits the infection to one more person.

Point 6b:

Given the objective of your paper, it also seems important to explicitly state that implementing a lockdown strategy at the beginning of the epidemic was critical but in the longterm for economic reasons may not be feasible, particularly given the density of the population and economic disparities in India. You make the following statement "Continuing a nationwide lockdown would not be feasible in the long term, and restrictions have to be eased in a phase-wise manner," but provide no references or further explanation for why it is not feasible and why restrictions should be phase-wise or examples. One of the most interesting/important aspects of understanding transmission in India is the population density and mobility patterns of the population - it would be helpful to expand on how your results apply to this or how future surveillance and studies should account for this. Why is knowing your results useful for future non-pharmaceutical interventions (NPIs) in India particularly when lockdown has historically shown the greatest reduction for this epidemic and the 1918 Flu and you're suggesting it's not feasible in India to maintain? Please edit your discussion/conclusion to address this.

Response:

From our statement, "Continuing a nationwide lockdown would not be feasible in the long term, and restrictions have to be eased in a phase-wise manner," we meant continuing stringent lockdown where almost everything was at halt from transportation to industry will not be feasible in long term. Also, we meant lockdown at “national” level would not be feasible, considering the heterogeneity in disease spread and response at the state level. We are in agreement that NPIs in the form of social or physical distancing helps in reducing spread. Results of this study also highlight the important role of NPIs in reducing the spread. We have modified discussion to clarify this. We also took your suggestion and updated the references for better clarity. The relevant text of the Discussion Section is provided below:

Therefore, in the post-lockdown era, it might be a challenge to maintain this path, and this may be the period where the absolute burden of the infected persons will be high.[29, 30] Also, there has been a disproportionately higher burden of serious infections, including those requiring intensive-care among individuals more than 60 years of age as compared to younger adults.[31] This, coupled with the higher prevalence of comorbid conditions (50%) in individuals over 60 years in India, may warrant a strategy tailored to this section of the population. [32] This also suggests that in addition to the identification of infection, it is imperative to shift the focus on mortality prevention. Containment strategies like lockdown have given us the much-needed opportunity to delay the peak and flatten the epidemic-curve. The time bought should be utilized to intensify the surveillance among ‘at-risk’ individuals and buttress the health infrastructure, including hospital beds with oxygen availability and critical care beds with ventilators and telemedicine. [33-35] 

At this juncture, an empirical question arises whether (despite showing the initial success) should the stringent lockdown be continued for a more extended period? Considering the undesired collateral effects of stringent restrictions on the economy and livelihoods of the general population; a nationwide lockdown may not be a feasible solution for a longer duration. Other NPIs (social distancing measures, wearing masks, legal enforcement to curtails the non-essential gatherings, etc.) should be enforced to compensate for the increased probability of random mixing. The decision on which NPI measure should be enforced should vary with the burden of active infections, emerging patterns of severity /mortality, and health system endurance and capacity to deal with such cases embedded in socio-economic and socio-cultural vulnerability. 

Another relevant observation in Indian COVID-19 context is that it does not look like an outbreak with similar intensity at the pan-country level. It seems to be a complex aggregation of several individual outbreaks occurring at different time points at different geographic locations. In principle, the magnitude of these outbreaks should be influenced by population density (outbreaks first started in areas where the population density is high), mobility patterns (higher number of cases were seen in places with better connectivity, i.e. international flights and domestic public transport systems) and the response of the healthcare system, all of which vary across different geographic locations. There is an urgent need for a real-time monitoring system that would take into consideration the disease burden (incidence and mortality), transmission parameters (reproduction number, doubling time and growth rate), existing health infrastructure (including bed capacity, human resources, etc.) and the vulnerability of other essential and frontline sectors.[36] This dynamic monitoring environment could serve as a sensitive tool to detect changes in the epidemiological pathways of COVID-19 and therefore, may facilitate the decision-making process on the nature and extent of NPI enforcement. This statement becomes more pertinent with the findings of our study, where we witness varying trajectories across the ten selected Indian states in response to the nationwide lockdown. Thus, logically the NPI enforcement should be tailored and customized according to the transmission parameters of smaller geographical areas, and hence the proposed monitoring system may play a pivotal role in this regard.

---

## [Editor Report · Decision Letter 2]

31 Aug 2020

Impact of COVID-19 epidemic curtailment strategies in selected Indian states: an analysis by reproduction number and doubling time with incidence modelling

PONE-D-20-14797R2

Dear Dr. Joshi,

We’re pleased to inform you that your manuscript has been judged scientifically suitable for publication and will be formally accepted for publication once it meets all outstanding technical requirements.

Kind regards,

Shinya Tsuzuki, MD, MSc

Academic Editor

PLOS ONE
---

## [Editor Report · Acceptance letter]

3 Sep 2020

PONE-D-20-14797R2 

Impact of COVID-19 epidemic curtailment strategies in selected Indian states: an analysis by reproduction number and doubling time with incidence modelling 

Dear Dr. Joshi:

I'm pleased to inform you that your manuscript has been deemed suitable for publication in PLOS ONE. Congratulations! Your manuscript is now with our production department. 

Kind regards, 

on behalf of

Dr. Shinya Tsuzuki 

Academic Editor

PLOS ONE